# Machine learning for predicting the diagnosis of tuberculous versus malignant pleural effusion: External validation and accuracy in two different settings

**Alberto Garcia-Zamalloa**[1]*, **Rafael Arnay**[2], **Iván Castilla-Rodriguez**[2], **Javier Mar**[3], **Jose Manuel Gonzalez-Cava**[2], **Oliver Ibarrondo**[4], **Iñaki Salegui**[5], **Juan Antonio De Miguel**[5], **Nekane Mugica**[5], **Borja Aguinagalde**[6], **Jon Zabaleta**[6], **Begoña Basauri**[7], **Marta Alonso**[8], **Nekane Azcue**[8], **Eva Gil**[9], **Irati Garmendia**[10], **Jorge Taboada**[4,11]

1 Respiratory and Pleural Diseases Group. Biogipuzkoa Health Research Institute. Internal Medicine Service, Osakidetza/Basque Health Service, Mendaro Hospital, Spain, 2 Departamento de Ingeniería Informática y de Sistemas, Universidad de La Laguna, Santa Cruz de Tenerife, Spain, 3 Epidemiology and Public Health Area, Economic Evaluation of Chronic Diseases Research Group, Biogipuzkoa Health Research Institute, Donostia, Spain, 4 Western Gipuzkoa Clinical Research Unit, Osakidetza/ Basque Health Service, Mendaro Hospital, Gipuzkoa, Spain, 5 Respiratory and Pleural Diseases Group. Biogipuzkoa Health Research Institute. Pneumology Service, Osakidetza/Basque Health Service, Donostia University Hospital, Gipuzkoa, Spain, 6 Respiratory and Pleural Diseases Group. Biogipuzkoa Health Research Institute. Thoracic Surgery Service, Osakidetza/Basque Health Service, Donostia University Hospital, Gipuzkoa, Spain, 7 Biochemistry Laboratory, Osakidetza/Basque Health Service, Mendaro Hospital, Gipuzkoa, Spain, 8 Microbiology Department, Respiratory Infection and Antimicrobial Resistance Group. Osakidetza/Basque Health Service, Biogipuzkoa Health Research Institute, Donostia University Hospital, Gipuzkoa, Spain, 9 Biochemistry Laboratory, Osakidetza/Basque Health Service, Donostia University Hospital, Gipuzkoa, Spain, 10 Oncology Area, Biogipuzkoa Health Research Institute, Gipuzkoa, Spain, 11 Preventive Medicine and Public Health Service, Bilbao-Basurto Integrated Organization, Bizkaia, Spain.

* alberto.garciazamalloa@gmail.com

## Abstract

### Objective

To perform an external validation of a previously reported machine learning (ML) approach for predicting the diagnosis of pleural tuberculosis.

### Patients and Methods

We defined two cohorts: a Training group, comprising 273 out of 1,220 effusions from our prospective study (2013–2022); and a Testing group, from a retrospective analysis of 360 effusions from 832 consecutive patients in Bajo Deba health district (1996–2012). All the effusions included were exudative and lymphocytic. In Training and Testing groups respectively, 49 and 104 cases were tuberculous, 143 and 92 were malignant, and 81 and 164 were diagnosed with "other diseases"; pre-test probabilities of pleural tuberculosis were 4% and 12.7%. Variables included were: age, pH, adenosine deaminase, glucose, protein, and lactate dehydrogenase levels, and

**Data availability statement:** All the data included in the study are fully available and uploaded in the Zenodo public repository: https://doi.org/10.5281/zenodo.15576280 Additionally we have deposited our laboratory protocols in protocols.io, in order to enhance the reproducibility of our results: dx.doi.org/10.17504/protocols.io.4r3l2pw53g1y/v1.

**Funding:** The author(s) received no specific funding for this work.

**Competing interests:** The authors have declared that no competing interest exist.

white cell counts (total and differential) in pleural fluid. We used two ML classifiers: binary (tuberculous and non-tuberculous), and three-class (tuberculous, malignant, and others); and compared them with Bayesian analysis.

## Results

The best binary classifier yielded a sensitivity of 88%, specificity of 98%, and accuracy of 95%. The best three-class classifier achieved the same accuracy and correctly classified 83% (77/92) of malignant cases. The ML models yielded higher positive predictive values than Bayesian analysis based on ADA > 40 U/l and lymphocyte percentage ≥ 50% (92%).

## Conclusions

This external validation confirms the good performance of the previously reported ML approach for predicting the diagnosis of pleural tuberculosis based on exudative and lymphocytic pleural effusions, and for discriminating the cases most likely to be malignant. Additionally, ML was more accurate than the Bayesian approach in our study.

## Introduction

Tuberculosis (TB) remains a major global public health problem. In 2024, the World Health Organization estimated that 10.8 million people worldwide had TB (95% uncertainty interval (UI): 10.1–11.7 million) and 1.25 million of them died of this disease through 2023 [1]. Tuberculous pleural effusion (TPE) is the second most prevalent manifestation of extrapulmonary TB, just after tuberculous lymphadenitis [2]. Additionally, TPE constitutes a paucibacillary manifestation of tuberculous disease, and the usefulness of various biomarkers in pleural fluid has been extensively analysed. The most promising biomarker is adenosine deaminase (ADA), as the measurement of this enzyme is cheap, easy and available in most laboratories, and it has shown uniformly high diagnostic performance through five consecutive meta-analyses with a mean sensitivity and specificity of 92–93% and 90–92% respectively, for a diagnostic cut-off point of 40 U/l [3–7].

The approach for diagnosing incident cases has traditionally been determined by the regional variation in overall TPE prevalence [8]; in relation to this, based on a Bayesian interpretation of its diagnostic accuracy, ADA in pleural fluid is considered optimal as a rule-in test in high TB prevalence settings and as a rule-out test in low prevalence scenarios [9,10]. More recently, Shaw et al. proposed an ADA level greater than 40 U/L for ruling in TPE diagnosis with a local TB global incidence of over 125 cases per 100,000 population, and ADA less than 30 U/L for ruling out TPE in low-incidence countries [11].

The progressively wider use of machine learning (ML) tools in medicine is providing researchers with new opportunities to improve diagnostic accuracy by including additional biomarkers. In 2021, our group published a comparison of different ML

algorithms for the diagnosis of TPE in a low prevalence scenario (3.8%) based on 230 exudative and lymphocytic pleural effusions diagnosed in Gipuzkoa Region from 2013 to 2020 [12]. The variables included in the predictive models were age, ADA and results from the routine analysis of pleural fluid including cell counts and biochemical tests. The best ML algorithm (support vector classifier, SVC, a particular type of support vector machine) achieved a sensitivity and specificity of 91% and 98% respectively; furthermore, compared with the Bayesian analysis of ADA level > 40 U/l plus lymphocyte percentage ≥ 50% in pleural fluid, the positive predictive value (PPV) increased from 42.4% to 70.5% in a 5% pre-test probability scenario. As the presumptive diagnosis of TPE has been traditionally driven by its prevalence, we needed to confirm that the implemented tool maintains its diagnostic properties in other scenarios featuring a different tuberculous prevalence.

The aim of this study was to perform an external validation of the aforementioned model by testing the trained ML algorithm in a different TPE prevalence setting, namely 360 pleural exudative and lymphocytic pleural effusions diagnosed in Bajo Deba Health District from 1996 to 2012. In a second step, we compared the diagnostic accuracy of the ML procedure and the classical Bayesian analysis system for TPE in both different clinical scenarios (Bajo Deba 1996–2012 and Gipuzkoa 2013–2022).

## Materials and methods

We pooled data from two groups of patients from Gipuzkoa (Basque Country, Spain) from January 1996 to the present. All the patients had been diagnosed with pleural effusion, undergone diagnostic thoracentesis, and had ADA level measured in the pleural fluid sample obtained. The first group, with which the ML models were trained (Training group), corresponded to patients from the whole region of Gipuzkoa, included in a previously published article regarding the diagnostic accuracy of ML for TPE [12], and for whom data were collected from January 2013 to December 2020. We extended this group with patients prospectively recruited across the region until December 2022. The second group, with which the ML models were tested (Testing group), comprised a retrospective cohort of patients from a specific area (Bajo Deba health district) for whom data were collected from January 1996 to December 2012 (partial results from 1998 to 2008 were published in 2012) [13]. The retrospective project counted with the permission of the local Ethics Committee of the Mendaro Hospital and the approval of the Western Gipuzkoa Clinical Research Commission; in the prospective project all patients gave written informed consent (the parents of a seventeen years old girl signed the unique informed consent for a minor) and the protocol was evaluated and approved by the Clinical Research Ethics Committee of Gipuzkoa (Record number 11/12). Data were accessed for research purposes from September to December 2012 within the retrospective project and from March 2013 to December 2022 for the prospective one.

The mean annual incidence rates of TB in Gipuzkoa from 2013 to 2022 and in Bajo Deba health district from 1996 to 2012 and in were 12.59 and 43.17 cases per 100,000 population respectively. Although a Tuberculosis Control Programme was not implemented in the Basque Country until 2003 [14], data recording has been highly reliable since 1995.

The following variables were included: age, pleural fluid ADA level, and pleural fluid routine test results: pH, glucose, protein, and lactate dehydrogenase (LDH) levels, and white blood cell counts (total and differential). It was not possible to include the red blood cell count in the analysis since it had not been routinely assessed in the referral hospital of Bajo Deba health district during the specified period (1996–2012). Further, to ensure consistency with the methodology employed in the 2013–2022 Gipuzkoa study, we considered only the first pleural fluid sample from each case. The databases were anonymized and authors did not have access to information that could identify individual participants.

Diagnostic criteria for tuberculous, malignant and parapneumonic pleural effusion applied to all the cases from the external validation/Testing group were as follows [15,16]:

- Confirmed tuberculous pleural effusion: positive culture or Xpert MTB/RIF assay in pleural fluid, pleural tissue or sputum.

- Probable tuberculous pleural effusion: granulomatous inflammation in pleural tissue and/or exudative and lymphocytic pleural effusion with ADA > 40 U/l and complete recovery with antituberculosis treatment.

- Confirmed malignant pleural effusion: malignant cells found in pleural biopsy tissue or pleural fluid.

- Paramalignant pleural effusion: cancer diagnosed de novo, no other cause of pleural effusion identified, absence of malignant cells in pleural fluid and pleural biopsy tissue, and parallel evolution of malignant disease and pleural effusion.

- Parapneumonic effusion: alveolar consolidation diagnosed by chest X-ray or computed tomography with ipsilateral pleural effusion, and total recovery of both with antibiotic treatment.

The rest of the cases in this group were diagnosed by well-defined clinical criteria, and all the cases in the Training group had gold standard quality diagnoses [12].

Overall, in the Training group (Gipuzkoa 2013–2022), 273 pleural effusions were exudative and lymphocytic and, of these, 49 were of tuberculous origin and 143 malignant. Regarding, the Testing group (Bajo Deba 1996–2012), 360 out of 832 episodes of pleural effusion were exudative and lymphocytic, and TB was diagnosed in 104 out of these 360 cases. Additionally, 92 cases were found to have a malignant aetiology.

The sample size for the Training group was computed by using a statistical program (Epidat 3.1). For a sensitivity of 95%, specificity of 90%, prevalence (pre-test probability) of 10%, significance level of 5%, power of 80%, and precision of 5%, the minimum sample size was 200 pleural effusions (12). Regarding the Testing group, we tried to include as many pleural effusions as possible due to the fact that the data recording was retrospective; we were able to record reliable data from 1996 to 2012.

In line with previous studies on pleural diseases since the 1990s, the term "prevalence" was employed to refer to the number of cases of a specific type of pleural effusion divided by the total number of pleural effusions studied in a given clinical setting over a known period of time. In this context, "prevalence" can be considered a synonym of "pre-test probability". Nonetheless, it should be noted that the term "pre-test probability" for ML specifically refers to the number of pleural fluid samples with a given diagnosis divided by the number of pleural fluid samples included in the study, not in the clinical scenario. Table 1 reports both of these variables; additionally, Supporting information (S1 Table and S2 Table) list the aetiology and diagnostic criteria met for all the cases included in the study.

Age and pleural fluid ADA level were by far the most relevant variables in our reference study (Training Group), and therefore, we compared the two populations as a function of these two variables and in a categorical way. Results of these comparisons are presented in Table 2.

**Table 1. Number of cases in the Training and Testing groups; pre-test probability for 1) local prevalence of tuberculous pleural effusion and 2) cases included in the machine learning analysis (that is, considering only the cases with exudative and lymphocytic effusion samples).**

| Group | GIPUZKOA 2013–2022 (Training group) | BAJO DEBA 1996–2012 (Testing group) |
|---|---|---|
| Cases with thoracentesis, (n) | 1220 | 832 |
| Exudative and lymphocytic pleural effusions included, (n) | 273 | 360 |
| Non-tuberculous cases (based on exudative and lymphocytic effusions), (n) | 224 | 256 |
| Cases of tuberculosis, (n) | 49 | 104 |
| Local pre-test probability or "prevalence" | 4% (49/1220) | 12.5% (104/832) |
| Machine learning pre-test probability or "prevalence" | 17.9% (49/273) | 28.8% (104/360) |
| Malignant cases, (n) | 143 | 92 |
| Other cases, (n) | 81 | 164 |

**Table 2. Comparative analysis of categorical variables age and ADA level in the reference (Training) and external validation (Testing) groups (considering $p < 0.05$ indicative of a statistically significant difference).**

| Diagnosis | | Tuberculosis | | | Malignancy | | | Others | | |
|---|---|---|---|---|---|---|---|---|---|---|
| Patient group | | Gipuzkoa 2013–2022 | Bajo Deba 1996–2012 | p value | Gipuzkoa 2013–2022 | Bajo Deba 1996–2012 | p value | Gipuzkoa 2013–2022 | Bajo Deba 1996–2012 | p value |
| Number of cases | | 49 | 104 | – | 143 | 92 | – | 81 | 164 | – |
| Age (years) | <30 | 7 (14.29%) | 57 (54.81%) | **<0.001** | 0 (0.00%) | 0 (0.00%) | 0.191 | 2 (2.47%) | 3 (1.83%) | 0.872 |
| | 30-60 | 22 (44.90%) | 22 (21.15%) | | 34 (23.78%) | 15 (16.30%) | | 19 (23.46%) | 36 (21.95%) | |
| | ≥60 | 20 (40.82%) | 25 (24.04%) | | 109 (76.22%) | 77 (83.70%) | | 60 (74.07%) | 125 (76.22%) | |
| ADA | <=40 U/l | 2 (4.08%) | 8 (7.69%) | 0.503 | 133 (93.01%) | 89 (96.74%) | 0.258 | 77 (95.06%) | 159 (96.95%) | 0.483 |
| | >40 U/l | 47 (95.92%) | 96 (92.31%) | | 10 (6.99%) | 3 (3.26%) | | 4 (4.94%) | 5 (3.05%) | |

As we can see, "age" and "ADA level" show similar values and behaviour in the two groups. The only statistically significant difference was found in the age of the two populations diagnosed with TPE: they were younger in the external validation or Testing group (Bajo Deba 1996–2012), and this is attributable to the TB incidence in the corresponding period being higher than in the Training group (Gipuzkoa 2013–2022), implying more TB bacilli circulating in the community and in turn a higher proportion of primary infections like cases of TPE in young people [17,18].

Along with this, Fig 1 shows the distribution of "tuberculous" and "others" samples as a function of ADA and age in both Training and Testing groups (recalling that all pleural fluid samples are exudative and lymphocytic, and hence, this distribution is the result of combining the two variables ADA > 40 U/l and lymphocyte percentage ≥ 50%: ADA 40 + LP 50). In this figure, there is a horizontal line representing the ADA level = 40 U/l. We can conclude that patients in both Training and Testing groups were evenly distributed, and hence, a comparison between them is valid.

Two distinct analyses were conducted. The primary analysis employed ML techniques to categorize samples into two groups: tuberculous and non-tuberculous. A secondary analysis utilized the same techniques to further differentiate malignant cases, resulting in three categories: tuberculous, malignant, and "others". During both the training and testing phases, the following variables were used: age, pleural fluid ADA level, and routine parameters derived from pleural fluid analysis (including pH, glucose, protein, and LDH levels, and total and differential white blood cell counts). In this study, the six classifiers employed were the same as those used in the previous study, and are among the most widely used types of ML classifier: multilayer perceptron, logistic regression, SVC, decision tree, K-nearest neighbours, and random forest. As in our previous study, we utilized the Python scikit-learn library to implement these classifiers. For each classifier, we explored a range of parameter values in conjunction with a 5-fold cross-validation approach.

Additionally, we compared positive and negative predictive values as a function of pre-test probability for the different trained ML models, as well as for a classification based on the ADA 40 + LP 50 criterion. The estimated positive predictive value (PPV) was calculated as a function of the pre-test probability (prevalence) using the sensitivity and specificity of each classifier obtained in the training dataset (Gipuzkoa):

$$(\text{sensitivity} * \text{prevalence}) / ((\text{sensitivity} * \text{prevalence}) + (1 - \text{specificity}) * (1 - \text{prevalence})).$$

The real PPVs in Gipuzkoa and Bajo Deba were calculated as the true positives divided by the sum of the true positives and false positives, obtained in each dataset:

$$\text{TP} / (\text{TP} + \text{FP})$$

Finally, we performed a three-class classification: tuberculous, malignant and others for comparability with our previous study.

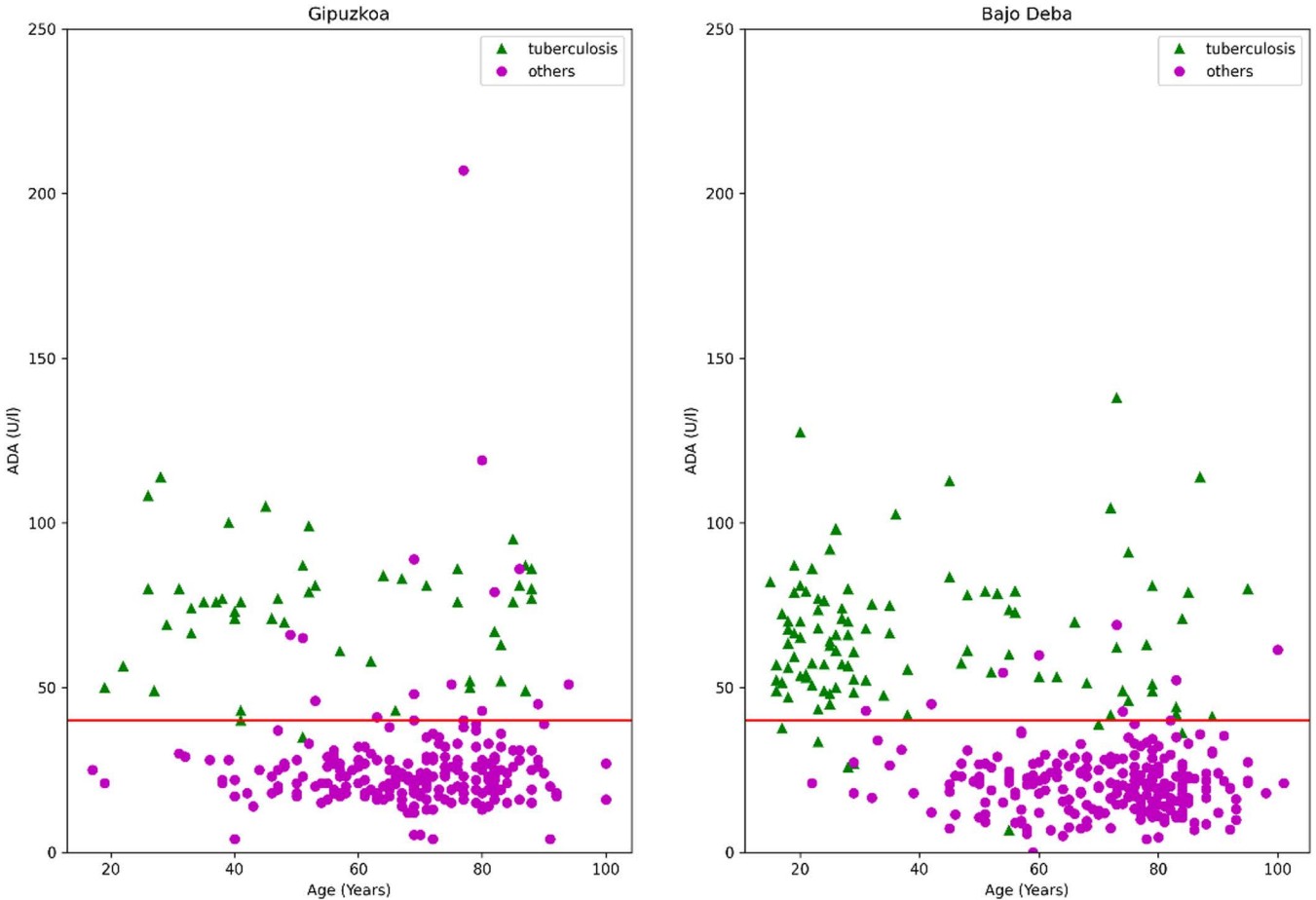

**Fig 1. Distribution of "Tuberculosis" and "others" samples in terms of ADA and age in Training (Gipuzkoa 2013-2022, left) and Testing (Bajo Deba 1996-2012, right) groups.**

## Results

Fig 2 shows the confusion matrices of the cross-validation results for all the ML models in the training set (Gipuzkoa), and the testing results (Bajo Deba) (0: others, 1: tuberculous; predicted values in columns, real values in rows).

Besides the classification results obtained with the ML models, we also conducted the same classification using the ADA 40 + LP 50 criterion to classify a sample as tuberculous. Using this criterion, we classified 210 cases of non-tuberculous disease as non-tuberculous (true negatives, TNs), 47 cases of TB as tuberculous (true positives, TPs), 14 cases of non-tuberculous disease as tuberculous (false positives, FPs) and 2 cases of TB as non-tuberculous disease (false negatives, FNs) in the Gipuzkoa dataset. On the other hand, in the Bajo Deba dataset, we obtained 248 TNs, 96 TPs, 8 FPs and 8 FNs.

Table 3 lists the metrics of accuracy, sensitivity, specificity, and PPV for each dataset for the different trained ML models and the ADA 40 + LP 50 criterion.

In Fig 3, the comparison of positive and negative predictive values as a function of pre-test probability is shown for the different trained ML models, as well as for a classification based on the ADA 40 + LP 50 criterion. The dashed vertical lines indicate the ML TPE pre-test probability points for the training (0.18) and testing (0.29) datasets (recall that the corresponding clinical scenario pre-test probabilities or "local prevalence of TPE" are 4% and 12.5% respectively).

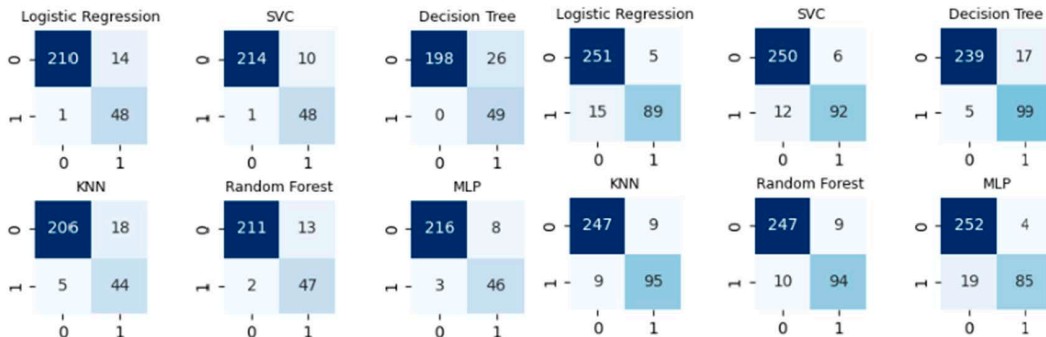

**Fig 2. Confusion matrices in cross-validation (Gipuzkoa, left) and in testing (Bajo Deba, right).**

**Table 3. Accuracy, sensitivity, specificity, and positive predictive value for each dataset for the trained machine learning models.**

| Classifier type | Gipuzkoa dataset | | | | Bajo Deba dataset | | | |
|---|---|---|---|---|---|---|---|---|
| | Accuracy | Sensitivity | Specificity | PPV | Accuracy | Sensitivity | Specificity | PPV |
| Logistic regression | 0.95 | 0.98 | 0.94 | 0.77 | 0.94 | 0.86 | 0.98 | 0.95 |
| Support vector machine | 0.96 | 0.98 | 0.96 | 0.83 | 0.95 | 0.88 | 0.98 | 0.94 |
| Decision tree | 0.9 | 1.0 | 0.88 | 0.65 | 0.94 | 0.95 | 0.93 | 0.85 |
| K-nearest neighbours | 0.92 | 0.9 | 0.92 | 0.71 | 0.95 | 0.91 | 0.96 | 0.91 |
| Random forest | 0.95 | 0.96 | 0.94 | 0.78 | 0.95 | 0.9 | 0.96 | 0.91 |
| Multilayer perceptron | 0.96 | 0.94 | 0.96 | 0.85 | 0.94 | 0.82 | 0.98 | 0.96 |
| ADA 40 + LP 50 | 0.94 | 0.96 | 0.94 | 0.77 | 0.96 | 0.92 | 0.97 | 0.92 |

PPV, positive predictive value; ADA 40 + LP 50, adenosine deaminase > 40 U/l and lymphocyte percentage ≥ 50%.

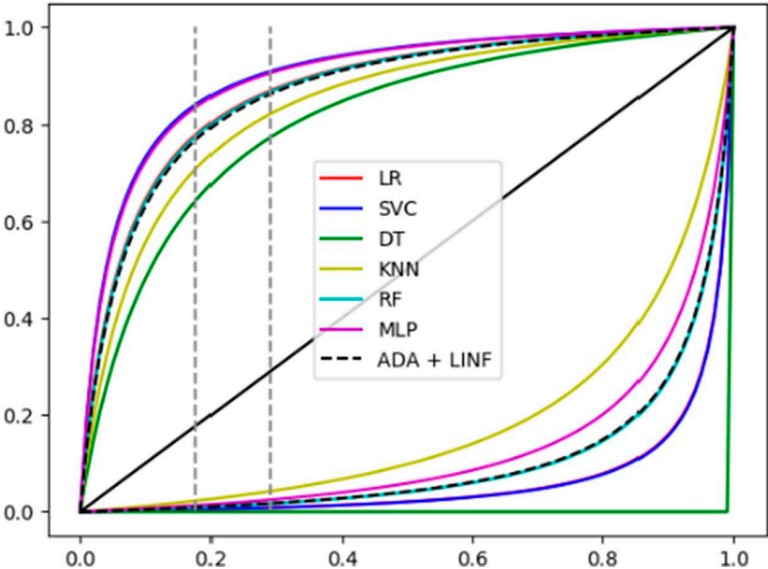

**Fig 3. Positive and negative predictive values as a function of pre-test probability.**

Table 4 compares the real and estimated PPVs obtained for each ML model as well as for a classification based on ADA 40 + LP 50, at the particular pre-test probabilities in the training (Gipuzkoa: 0.18) and testing (Bajo Deba: 0.29) datasets.

It can be observed that the PPV from the ML models exceeds that based on the ADA 40 + LP 50 criterion in several cases, both for the Gipuzkoa and Bajo Deba datasets. Additionally, the estimated PPVs underestimate the real values obtained in the Bajo Deba dataset for all ML models as well as for the ADA 40 + LP 50-based classifier.

Fig 4 shows the confusion matrices of the classification results in Gipuzkoa (left) and Bajo Deba (right), respectively (0: tuberculous, 1: malignant, and 2: others; predicted values in columns, real values in rows).

## Discussion

In this study, we have performed an external validation of our results from a prospective study that assessed the usefulness of ML for predicting the diagnosis of TPE, going beyond the ADA 40 + LP 50 criterion, in a low prevalence setting and through the period 2013–2022 [12]. We decided to conduct the study in a different prevalence scenario and we found some additional interesting behaviours of the ML classifiers in this context.

TPE constitutes a type 4 hypersensitivity reaction to mycobacterial antigens, and it is not certain whether this immunogenic response needs live mycobacteria. There is a well-proven initial invasion of polymorphonuclear leukocytes in the pleural fluid, followed by macrophages, and finally, almost all TPE fluids become lymphocytic [19,20]. An ideal diagnostic tool to confirm TPE should be able to detect *Mycobacterium tuberculosis* or any of its specific subunits or components

**Table 4. Real and estimated positive predictive values in the training and testing datasets.**

| Classifier type | Gipuzkoa dataset | | Bajo Deba dataset | |
| --- | --- | --- | --- | --- |
| | Estimated | Real | Estimated | Real |
| Logistic regression | 0.78 | 0.77 | 0.87 | 0.95 |
| Support vector machine | 0.84 | 0.83 | 0.91 | 0.94 |
| Decision tree | 0.64 | 0.65 | 0.77 | 0.85 |
| K-nearest neighbours | 0.71 | 0.71 | 0.82 | 0.91 |
| Random forest | 0.77 | 0.78 | 0.87 | 0.91 |
| Multilayer perceptron | 0.83 | 0.85 | 0.91 | 0.96 |
| ADA 40 + LP50 | 0.77 | 0.77 | 0.86 | 0.92 |

ADA 40 + LP 50: adenosine deaminase > 40 U/l and lymphocyte percentage ≥ 50%.

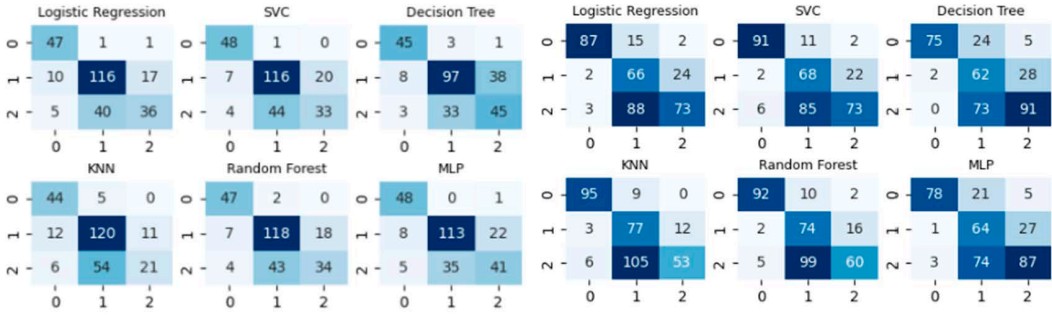

**Fig 4. Confusion matrices for each classifier in Gipuzkoa (left) and Bajo Deba (right) (0: tuberculous, 1: malignant, 2: others; predicted values in columns, real values in rows).**

                                                  8 / 12

directly, and optimally in pleural fluid through diagnostic thoracentesis; but due to its paucibacillary nature, current microbiological and nucleic acid amplification tests show low sensitivity: <5% with acid-fast bacilli smear in pleural fluid, 30–50% with mycobacterial culture and 21–47% with a nucleic acid amplification test compared to a composite reference standard [21]. Pleural biopsy (closed or guided by video-thoracoscopy) is more sensitive but also more invasive, and hence, it is not suitable in all clinical scenarios; further, *Mycobacterium tuberculosis* grows in pleural tissue culture in 75% of cases, the others being diagnosed based on the histological finding of granuloma in pleural tissue, which shows a specificity of 95% but gives no information regarding microbiological characteristics or drug resistance [22].

Due to all the above, great efforts have been made over recent decades to obtain a surrogate diagnostic tool for TPE, pleural fluid biomarkers being the most extensively explored. They provide a probability for ruling in and/or ruling out the disease, and among them, ADA is the most cost-effective, it is standardized and it has a universally accepted diagnostic cut-off value (40 U/l) [12]. Its combination with lymphocyte percentage increases ADA's specificity [23], as we also confirmed in our first study from 1998 to 2008 [13]. In our second study [12], we included both variables in an ML approach. This third study is an external validation of the latter, and we performed it in a higher TPE prevalence scenario.

Using the ADA 40+LP 50 criterion, specificity decreases when moving from a high- to a low-incidence scenario. This is because, proportionally, there are more malignant samples with ADA>40 U/l in low-incidence scenarios, which increases the number of false positives (8 classified erroneously as tuberculous out of a total of 256 non-tuberculous samples in the case of Bajo Deba, compared to 14 classified erroneously as tuberculous out of a total of 224 non-tuberculous samples in the case of Gipuzkoa).

Nonetheless, some ML models show a less marked decrease in specificity when transitioning from the high- to the low-incidence scenario. For example, the SVC achieved sensitivities and specificities of 98% and 96% in Gipuzkoa and 88% and 98% in Bajo Deba, while the logistic regression classifier provided sensitivities and specificities, respectively, of 98% and 94% in Gipuzkoa and 86% and 98% in Bajo Deba. This is because these models rely not only on the ADA 40+LP 50 criterion but also a combination of other features, each with its relative influence, ADA and age being the most important ones.

Both the ADA 40+LP 50 criterion and the ML models provide lower sensitivity in Gipuzkoa than in Bajo Deba health district. This can be explained by two factors. Firstly, there were quite a few cases of TB in Bajo Deba with ADA below 40 U/l (in the first pleural fluid sample), while there are virtually no such cases in Gipuzkoa, causing both the ML models and the ADA>40 criterion to fail to generalize correctly in the Bajo Deba dataset. Secondly, and this affects only the ML models, the patterns of ADA and age, the two most discriminatory variables, are not the same in the Gipuzkoa and Bajo Deba datasets. Fig 5 shows the tuberculous, malignant, and other cases as a function of ADA and age in Gipuzkoa (left) and in Bajo Deba (right). False negatives produced by the SVC in the Bajo Deba dataset have also been marked with red crosses. As can be seen, there was much more TB circulating in the young population in the Bajo Deba dataset (right) than in Gipuzkoa (left), and although the ML models have not seen this pattern in the training set, they are capable of generalizing well, since there were no non-tuberculous samples in those ranges of ADA and age in the Gipuzkoa dataset either. Nonetheless, there are ranges of ADA slightly above 40 U/l and ages around 80 years, where most samples in the Gipuzkoa dataset are classified as "malignant" or "other", while in the Bajo Deba dataset, there are tuberculous samples, and hence, most of the trained ML models fail to generalize well.

Comparing ML and Bayesian analysis, it can be observed that the PPV of the ML models exceeds that of ADA 40+LP 50 in several cases, for both the Gipuzkoa and Bajo Deba datasets. Additionally, the estimated PPVs underestimate the real values obtained in the Bajo Deba dataset for all ML models as well as for the ADA 40+LP 50-based classifier (Fig 3 and Table 4). This indicates that the "tuberculous" and "others" classes are more readily separable in the Bajo Deba than in the Gipuzkoa dataset. This is primarily attributable to the fact that there are fewer malignant cases with high ADA in Bajo Deba than in Gipuzkoa. Consequently, the projected PPV for a classifier with a sensitivity and specificity obtained in Gipuzkoa falls short of the real PPV obtained in Bajo Deba for that classifier.

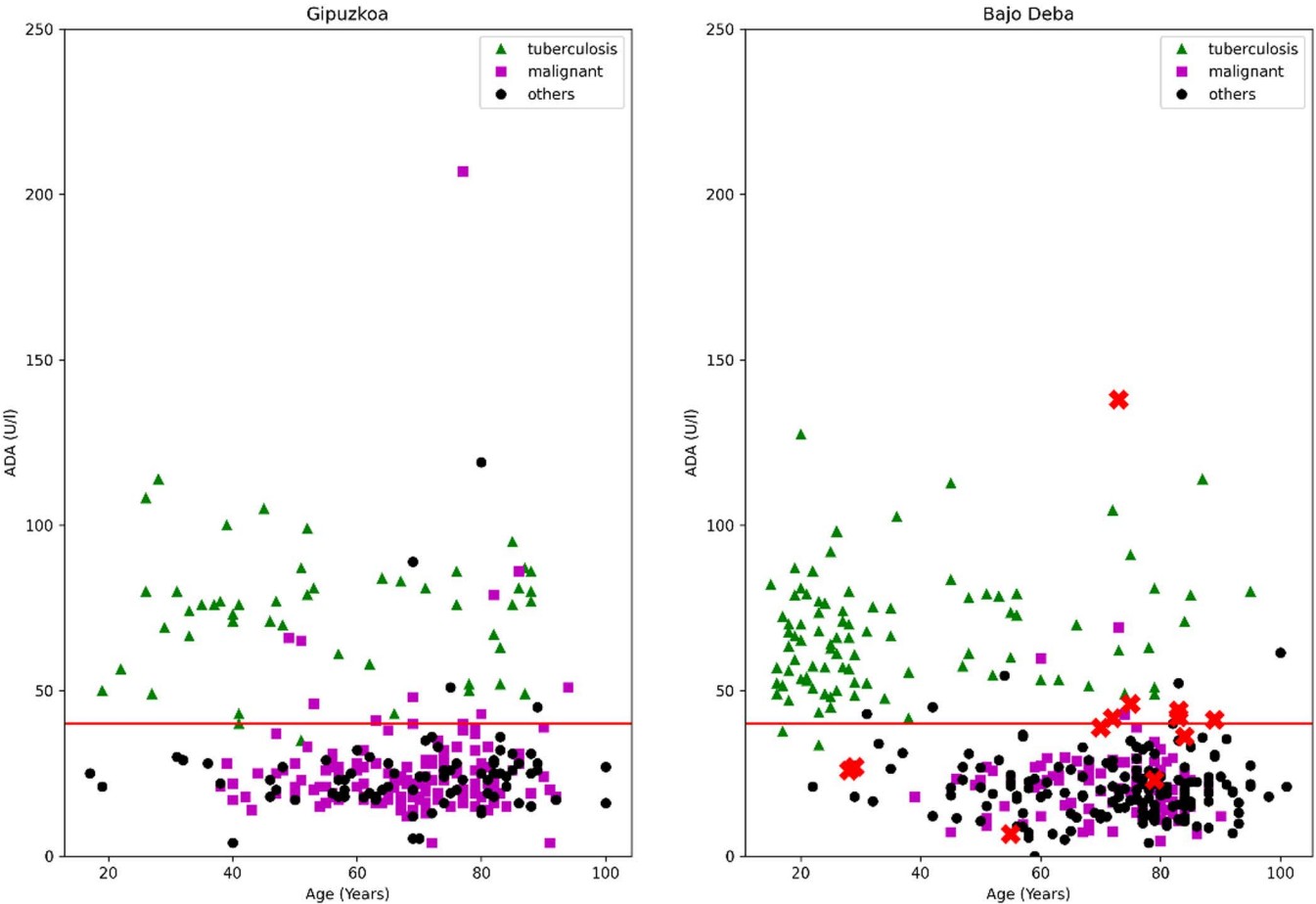

**Fig 5. Tuberculous, malignant, and other cases as a function of adenosine deaminase level and age in Gipuzkoa (left) and in Bajo Deba (right).** False negatives produced by the support vector classifier in the Bajo Deba dataset are marked with red crosses.

Concerning the results of the three-class classification (tuberculous/malignant/others), as can be observed, practically all three-class classifiers provide sensitivity/specificity in the classification of tuberculous cases against the others similar to that of the two-class classifiers. For example, the Random Forest classifier correctly detects 47 cases of TPE out of a total of 49 in Gipuzkoa and 92 out of a total of 104 in Bajo Deba, but at the same time, it is capable of correctly classifying 118 malignant cases out of a total of 143 malignant cases in Gipuzkoa, as well as correctly classifying 74 malignant cases out of a total of 92 malignant cases in Bajo Deba (Fig 4).

This study has some limitations. First, the analysis of the testing dataset is retrospective, it has been performed in a single centre and the quality of diagnosis is lower than in the training dataset. Second, the incidence rate of TB is higher in the testing group, but at the same time, this has provided an opportunity to analyse the behaviour of the ML classifiers in different prevalence scenarios of a transmissible disease. Moreover, the number of cases included in the study, the homogeneity of the groups and the comparable behaviour of the two most important variables are positive features, enhancing the validity of our study's findings. Our Machine Learning model is freely available as an app (at https://pleurapp.ispana.es/) to help other physicians or thoracic surgeons apply this approach when dealing with exudative and lymphocytic pleural effusions.

## Conclusions

With this external validation of our previously reported model, we confirm that an ML approach combining ADA with age and routine pleural fluid parameters is suitable for predicting the diagnosis of pleural TB in any prevalence scenario, and secondarily, for discriminating the cases most likely to be malignant amongst exudative and lymphocytic non-tuberculous effusions. Additionally, we conclude that ML is more accurate than Bayesian analysis in different prevalence scenarios, although the epidemiological behaviour of transmissible diseases (e.g., the relationship between incidence rate and patient age) is a challenge to be tackled in developing ML approaches.

## Supporting information

**S1 Table. Number of cases by diagnosis obtained in the Training and Testing groups.**
(DOCX)

**S2 Table. Diagnostic criteria met by tuberculous and malignant cases (listing each case just once under the highest quality criterion among all the criteria met).**
(DOCX)

## Author contributions

**Conceptualization:** Alberto Garcia-Zamalloa, Rafael Arnay, Begona Basauri, Jorge Taboada.

**Data curation:** Alberto Garcia-Zamalloa, Iñaki Salegui, Juan Antonio De Miguel, Nekane Mugica, Borja Aguinagalde, Jon Zabaleta, Marta Alonso, Nekane Azcue, Eva Gil.

**Formal analysis:** Rafael Arnay, Ivan Castilla-Rodriguez, Javier Mar, Jose Manuel Gonzalez-Cava, Oliver Ibarrondo, Irati Garmendia.

**Investigation:** Alberto Garcia-Zamalloa, Juan Antonio De Miguel, Borja Aguinagalde, Irati Garmendia.

**Methodology:** Rafael Arnay, Ivan Castilla-Rodriguez, Javier Mar, Jose Manuel Gonzalez-Cava, Oliver Ibarrondo.

**Software:** Rafael Arnay, Ivan Castilla-Rodriguez.

**Validation:** Rafael Arnay, Ivan Castilla-Rodriguez.

**Writing – original draft:** Alberto Garcia-Zamalloa, Rafael Arnay, Ivan Castilla-Rodriguez, Javier Mar.

**Writing – review & editing:** Alberto Garcia-Zamalloa, Rafael Arnay, Ivan Castilla-Rodriguez, Javier Mar.

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
