## [Decision Letter · Decision Letter 0]

15 May 2025

PONE-D-25-09667Machine learning for predicting the diagnosis of pleural tuberculosis: external validation and accuracy in two different settings.PLOS ONE

Dear Dr. Garcia-Zamalloa,

Thank you for submitting your manuscript to PLOS ONE. After careful consideration, we feel that it has merit but does not fully meet PLOS ONE’s publication criteria as it currently stands. Therefore, we invite you to submit a revised version of the manuscript that addresses the points raised during the review process.

We look forward to receiving your revised manuscript.

Kind regards,

Guocan Yu

Academic Editor

PLOS ONE

3. In the online submission form, you indicated that [All the data are included in a database in our Department, and they cannot be shared publicly, but we have no problem to share all them for everyone who ask for it and who meet the criteria for access to confidential data.].

Reviewers' comments:

Reviewer's Responses to Questions

**Comments to the Author**

1. Is the manuscript technically sound, and do the data support the conclusions?

Reviewer #1: Yes

Reviewer #2: Yes

2. Has the statistical analysis been performed appropriately and rigorously? 

Reviewer #1: Yes

Reviewer #2: Yes

3. Have the authors made all data underlying the findings in their manuscript fully available?

Reviewer #1: Yes

Reviewer #2: Yes

4. Is the manuscript presented in an intelligible fashion and written in standard English?

Reviewer #1: Yes

Reviewer #2: Yes

5. Review Comments to the Author

Reviewer #1: Peer Review for Manuscript PONE-D-25-09667

Date: 9-April-2025

“Machine learning for predicting the diagnosis of pleural tuberculosis: external validation and accuracy in two different settings”

These are my comments of the peer review for the manuscript requested.

My general comments:

In general, the authors presented a well-conducted of external validation study. This research was developed, based on a authors’ recent publication of development prospective cohort model (reported in 2021) of Adenosine deaminase (ADA) for pleural tuberculosis in low tuberculosis (TB) prevalence. The training cohort showed promising results, providing a good rationale for an external validation study. The analysis is very good; however, I think there is room for improvement in writing. To my own perception, I think the structure and presentation of this article writing are still not smooth.

The plus point of this external validation is that the developed pleural TB model (TPE) was tested in a completely different cohort with different TB/TPE incidence. In this study, although test cohort showed a bit lower accuracy, predictive values, than the Train study cohort. I think these real-life data are highly appreciated, demonstrating the applicability of the machine learning TPE models. This highlights the real-life data testing in the external validation cohort.

In clinical practice, tuberculous pleural effusion (TPE) is difficult for diagnosis, particularly in immunocompromised patients and/or limited resource countries with high TB burden. Hence, I raise a clinical question regarding the immune status of the study participants (training and test cohorts). Because HIV co-infection prevalence is highly significant among patients with TPE, do authors have any data about HIV infection or immune testing (CD4 cell counts…etc) in these cohorts?

In additionally, a short justification for sample size calculation in external validation cohorts enhances the validity of the study as well.

My specific comments:

1. Title: The title fully describes study aim and objectives.

2. Abstract: The abstract is well written. However, I also think one minor point for amendment, as below:

Lines 45 to 47:

The authors presented a sample size of the Test cohort with 832 consecutive patients in Bajo Deba health district (1996-2012), but did not show the sample size of the Train cohort from a prospective cohort study (2013-2020). To be consistent and transparent in data, the sample size (how many patients in Train cohort?) should be described (stated) in this section as well.

3. Introduction:

There is room for improvement in the introduction section, as follows:

Line 67: 95% UI needs to be fully written, as readers are not familiar with this term-UI.

Lines 92 to 94: “The model is freely 92 available as an app (at https://pleurapp.ispana.es/) to help other physicians or thoracic surgeons apply this approach when dealing with exudative and lymphocytic pleural effusions”

This information should not be placed in the introduction because it was not connected with the flow of the main idea discussed. This can be relocated to the discussion as appropriate.

Lines 99 to 100 in the introduction section:

“we compared the 99 diagnostic accuracy of the ML procedure and the classical Bayesian analysis system for TPE in both 100 different clinical scenarios (Bajo Deba 1996-2012 and Gipuzkoa 2013-2022)”.

The authors mentioned two study cohorts to be modelled without brief introduction before. Hence, I recommend a brief introduction (1-2 sentences) of these 2 cohorts in the previous paragraphs.

4. Materials and Methods and Results

There is room for Materials and Methods section for improvement, regarding data presentation.

4.1 I can understand that the authors aimed to place emphasis on the Test cohort (external validation), so proactively present the Test study cohort as the first group, while the Train study cohort as the second group. Intuitively, this style of data presentation brings readers (like me) to a certain level of confusion and needs to reread and rethink. Therefore, I recommend that the authors present data as routine to characterize the first group = Train cohort, and second group = external validation cohort.

This order of presentation should be consistent throughout the manuscript text and Tables. In the Tables 1 and 2, the authors first present: Test to Train cohorts (in sequence);

then in Tables 3 and 4: Train to Test cohorts (in sequence)

The transition in data presentation will make readers (eg, like my case) confused and take some time to reread, rethink the study data.

Therefore, I highly recommend a consistent presentation style of study data, I prefer training to testing cohort data presentation (from left to right), as the conventional way.

4.2 In Lines 213 to 218, the authors do not need to describe detailed data about confusion matrices of all machine learning models one by one, because readers can track all these information in Figures 2 and 4 presented. Only salient features from these data should be stated in the manuscript text.

4.3 Lines 237-239 and Lines 242-245

“The estimated PPV area is calculated as a function of the pre-test probability (prevalence) using the

sensitivity and specificity of each classifier obtained in the training dataset (Gipuzkoa):

(sensitivity * prevalence) / ((sensitivity * prevalence) + (1 - specificity) * (1 - prevalence)).”

“The real PPVs in Gipuzkoa and Bajo Deba are calculated as the true positives divided by the sum of the true positives and false positives, obtained in each dataset: TP / (TP + FP)”

The formulas should be relocated in the Methods section, as it is more appropriate.

5. Discussion:

The discussion is comprehensively discussed and well written. From this study, the machine learning models outperformed the Bayesian modelling, as shown in a different study setting with different prevalence of TPE and malignancy.

Conclusion:

I think this is a great study, and minor amendments are suggested to make it more comprehensible for readers. I agree that this study is appropriate for publication.

Many thanks,

Best regards,

Reviewer #2: Overall well-written.

See attached DOCX file for some reorganizing suggestions.

Some of the Results have been included in the Methods section.

The incidence of TB between the two groups is not so significant.

6. PLOS authors have the option to publish the peer review history of their article (what does this mean? ). If published, this will include your full peer review and any attached files.

**Do you want your identity to be public for this peer review?** For information about this choice, including consent withdrawal, please see our Privacy Policy .

Reviewer #1: **Yes: ** Nguyen Tat Thanh (MD, PhD)

Reviewer #2: No

---

## [Author Response · Author response to Decision Letter 1]

4 Jun 2025

Reviewer #1: Peer Review for Manuscript PONE-D-25-09667

Date: 9-April-2025

“Machine learning for predicting the diagnosis of pleural tuberculosis: external validation and accuracy in two different settings”

- (REVIEWER): These are my comments of the peer review for the manuscript requested.

My general comments:

In general, the authors presented a well-conducted of external validation study. This research was developed, based on a authors’ recent publication of development prospective cohort model (reported in 2021) of Adenosine deaminase (ADA) for pleural tuberculosis in low tuberculosis (TB) prevalence. The training cohort showed promising results, providing a good rationale for an external validation study. The analysis is very good; however, I think there is room for improvement in writing. To my own perception, I think the structure and presentation of this article writing are still not smooth.

* (AUTHORS) Thank you very much for your general suggestion about the structure of the work. We have thoroughly followed your comments and, as a result, we think that we have improved the overall structure of the paper. Particularly, we have presented the training results first and then the testing results. Besides, we have improved the Material and Methods section,

- (R) The plus point of this external validation is that the developed pleural TB model (TPE) was tested in a completely different cohort with different TB/TPE incidence. In this study, although test cohort showed a bit lower accuracy, predictive values, than the Train study cohort. I think these real-life data are highly appreciated, demonstrating the applicability of the machine learning TPE models. This highlights the real-life data testing in the external validation cohort.

In clinical practice, tuberculous pleural effusion (TPE) is difficult for diagnosis, particularly in immunocompromised patients and/or limited resource countries with high TB burden. Hence, I raise a clinical question regarding the immune status of the study participants (training and test cohorts). Because HIV co-infection prevalence is highly significant among patients with TPE, do authors have any data about HIV infection or immune testing (CD4 cell counts…etc) in these cohorts?

* (A) Thank you for your question. Indeed, tuberculosis is more prevalent amongst patients coinfected with HIV, but in this sense we must state that:

- Screening for Human Immunodeficiencty virus (HIV) was performed in all patients diagnosed with any form of tuberculosis in the Gipuzkoa Region following the guidelines of the Tuberculosis Control Program implemented in the Basque Country since 2003.

- There were no cases of TPE coinfected with HIV in our series from 2013 to 2022 in Gipuzkoa Region, namely the Training Cohort. Only three patients diagnosed with HIV infection developed pleural effusion through this period, and it was malignant in all cases, as we reported in our prospective project (1). The absence of cases of TPE amongst HIV coinfected patients would have probably been due to the widespread antiretroviral treatment.

- Unfortunately, regarding the Testing Cohort in Bajo Deba Health District from 1996 to 2012, patients coinfected with HIV were attended and followed in the Regional Donostia University Hospital. This cohort was retrospective and we only have the information stored at the Bajo Deba Health District Hospital.

- Nevertheless, and as pointed out in our first report from 1998 to 2008 (2), ADA accuracy is known to be equally reliable in HIV-positive patients with TPE, even in those with low CD4 T-cell count (3,4), and even in renal transplant recipients (5).

o 1) Garcia-Zamalloa A, Vicente D, Arnay R, Arrospide A, Taboada J, Castilla-Rodriguez I, et al. (2021) Diagnostic accuracy of adenosinedeaminase for pleural tuberculosis in a low prevalence setting: A machine learning approach within a 7-year prospective multi-center study. PLoS ONE 16(11): e0259203. https://doi.org/10.1371/journal.pone.0259203

o 2) Garcia-Zamalloa A, et al. (2012) Diagnostic accuracy of adenosine deaminase and lymphocyte proportion in pleural fluid for tuberculous pleurisy in different prevalence scenarios. PLoSONE 7 (6): e38729.

o 3) Riantawan P, et al. (1999) Diagnostic value of pleural fluid adenosine deaminase in tuberculous pleuritis with reference to HIV coinfection and a Bayesian analysis. Chest 116: 97-103.

o 4) Baba K, et al. (2008) Adenosine deaminase activity is a sensitive marker for the diagnosis of tuberculous pleuritis in patients with low CD4 counts. PLoSONE 3 (7): e2788.

o 5) Krenke R, et al. (2010) Use of pleural fluid levels of adenosine deaminase and interferon gamma in the diagnosis of tuberculous pleuritis. Curr Opin Pulm Med 16: 367-375.

- (R) In additionally, a short justification for sample size calculation in external validation cohorts enhances the validity of the study as well.

* (A) Thank you very much for the suggestion. We have included this information in the manuscript.

Besides, we must state that our intention was to include as many pleural effusions as possible in the Testing Cohort, due to the fact that it was retrospective (Bajo Deba 1996-2012).

Nevertheless, as expressed in our report from 2021 (1), calculation of minimal sample size was set to 200 patients

o 1) Garcia-Zamalloa A, Vicente D, Arnay R, Arrospide A, Taboada J, Castilla-Rodriguez I, et al. (2021) Diagnostic accuracy of adenosinedeaminase for pleural tuberculosis in a low prevalence setting: A machine learning approach within a 7-year prospective multi-center study. PLoS ONE 16(11): e0259203. https://doi.org/10.1371/journal.pone.0259203

- (R) My specific comments:

1. Title: The title fully describes study aim and objectives.

* (A) We finally decided to modify the title by following the Reviewer 2´s suggestion: “Machine learning for predicting the diagnosis of tuberculous versus malignant pleural effusion: external validation and accuracy in two different settings”. Thank you.

2. Abstract: The abstract is well written. However, I also think one minor point for amendment, as below:

Lines 45 to 47:

The authors presented a sample size of the Test cohort with 832 consecutive patients in Bajo Deba health district (1996-2012), but did not show the sample size of the Train cohort from a prospective cohort study (2013-2020). To be consistent and transparent in data, the sample size (how many patients in Train cohort?) should be described (stated) in this section as well.

* (A) Thank you for the suggestion. We do find it very reasonable. We have included the number of pleural effusions of the Training cohort from 2013 to 2020, subsequently extended to 2022.

3. Introduction:

There is room for improvement in the introduction section, as follows:

Line 67: 95% UI needs to be fully written, as readers are not familiar with this term-UI.

* (A) Thank you for the suggestion. We have modified it.

- (R) Lines 92 to 94: “The model is freely available as an app (at https://pleurapp.ispana.es/) to help other physicians or thoracic surgeons apply this approach when dealing with exudative and lymphocytic pleural effusions”

This information should not be placed in the introduction because it was not connected with the flow of the main idea discussed. This can be relocated to the discussion as appropriate.

* (A) Thank you very much for your suggestion. We have moved it into the Discussion chapter as last paragraph.

- (R) Lines 99 to 100 in the introduction section:

“we compared the 99 diagnostic accuracy of the ML procedure and the classical Bayesian analysis system for TPE in both 100 different clinical scenarios (Bajo Deba 1996-2012 and Gipuzkoa 2013-2022)”.

The authors mentioned two study cohorts to be modelled without brief introduction before. Hence, I recommend a brief introduction (1-2 sentences) of these 2 cohorts in the previous paragraphs.

* (A) Thank you for the suggestion. We have included a brief exposition regarding the pleural effusions included in the two cohorts and the two different prevalence settings. Following the amendments of Reviewer 2, we also changed the term “higher prevalence setting” to “different prevalence setting”.

4. Materials and Methods and Results

There is room for Materials and Methods section for improvement, regarding data presentation.

4.1 I can understand that the authors aimed to place emphasis on the Test cohort (external validation), so proactively present the Test study cohort as the first group, while the Train study cohort as the second group. Intuitively, this style of data presentation brings readers (like me) to a certain level of confusion and needs to reread and rethink. Therefore, I recommend that the authors present data as routine to characterize the first group = Train cohort, and second group = external validation cohort.

This order of presentation should be consistent throughout the manuscript text and Tables. In the Tables 1 and 2, the authors first present: Test to Train cohorts (in sequence);

then in Tables 3 and 4: Train to Test cohorts (in sequence)

The transition in data presentation will make readers (eg, like my case) confused and take some time to reread, rethink the study data.

Therefore, I highly recommend a consistent presentation style of study data, I prefer training to testing cohort data presentation (from left to right), as the conventional way.

* (A) Thank you for the recommendation. We have restructured the text and tables to introduce Gipuzkoa (Training) first and then Bajo Deba (Testing).

4.2 In Lines 213 to 218, the authors do not need to describe detailed data about confusion matrices of all machine learning models one by one, because readers can track all these information in Figures 2 and 4 presented. Only salient features from these data should be stated in the manuscript text.

* (A) Thank you for the suggestion. We have added a clarification at the beginning of the paragraph to emphasize that we refer to the TP, FP, TN and FN values of a classification method simply consisting of using the ADA 40 + LP 50 criterion. The aim of this paragraph is to present a comparison with the results obtained by ML models shown in Figures 2 and 4.

4.3 Lines 237-239 and Lines 242-245

“The estimated PPV area is calculated as a function of the pre-test probability (prevalence) using the

sensitivity and specificity of each classifier obtained in the training dataset (Gipuzkoa):

(sensitivity * prevalence) / ((sensitivity * prevalence) + (1 - specificity) * (1 - prevalence)).”

“The real PPVs in Gipuzkoa and Bajo Deba are calculated as the true positives divided by the sum of the true positives and false positives, obtained in each dataset: TP / (TP + FP)”

The formulas should be relocated in the Methods section, as it is more appropriate.

* (A) Following your recommendation, we have modified the Material and Methods section to introduce the comparative analysis of ML and Bayesian analysis for estimating positive and negative predictive values as a function of pre-test probability. We have placed the mentioned formulas in this section.

5. Discussion:

The discussion is comprehensively discussed and well written. From this study, the machine learning models outperformed the Bayesian modelling, as shown in a different study setting with different prevalence of TPE and malignancy.

Conclusion:

I think this is a great study, and minor amendments are suggested to make it more comprehensible for readers. I agree that this study is appropriate for publication.

* (A) Many thanks,

Best regards,

- Reviewer #2: Overall well-written.

See attached DOCX file for some reorganizing suggestions.

Some of the Results have been included in the Methods section.

* (A) Thank you for the amendment. It is true that in the Material and Methods section it is shown a comparative analysis of our data. However, the aim of this analysis is to show that there are no statistically significant differences between both data sets. We prefer to see this analysis as part of the methodology that we followed to validate our Materials (Data), in order to proceed to train and test ML models with this data, rather than threat it as a result by itself. Also, following some suggestions of Reviewer 1, we have expanded the Material and Methods chapter with the methodology followed to train and test the ML models and the comparative analysis of ML and Bayesian analysis for estimating positive and negative predictive values as a function of pre-test probability

- (R) The incidence of TB between the two groups is not so significant.

* (A) Thank you for the suggestion. We modified the term “higher” prevalence for “different” prevalence.

We also changed the title following Reviewer 2’s suggestion and we made some corrections to keep a consistent order in the presentation of the Training and Testing results (in this order).

---

## [Decision Letter · Decision Letter 1]

21 Jul 2025

Machine learning for predicting the diagnosis of tuberculous versus malignant pleural effusion: external validation and accuracy in two different settings.

PONE-D-25-09667R1

Dear Dr. Alberto Garcia-Zamalloa,

We’re pleased to inform you that your manuscript has been judged scientifically suitable for publication and will be formally accepted for publication once it meets all outstanding technical requirements.

Kind regards,

Guocan Yu

Academic Editor

PLOS ONE

Additional Editor Comments (optional):

Reviewers' comments:

Reviewer's Responses to Questions

**Comments to the Author**

1. If the authors have adequately addressed your comments raised in a previous round of review and you feel that this manuscript is now acceptable for publication, you may indicate that here to bypass the “Comments to the Author” section, enter your conflict of interest statement in the “Confidential to Editor” section, and submit your "Accept" recommendation.

Reviewer #1: All comments have been addressed

Reviewer #3: All comments have been addressed

2. Is the manuscript technically sound, and do the data support the conclusions?

Reviewer #1: Yes

Reviewer #3: Yes

3. Has the statistical analysis been performed appropriately and rigorously? 

Reviewer #1: Yes

Reviewer #3: Yes

4. Have the authors made all data underlying the findings in their manuscript fully available?

Reviewer #1: Yes

Reviewer #3: Yes

5. Is the manuscript presented in an intelligible fashion and written in standard English?

Reviewer #1: Yes

Reviewer #3: Yes

6. Review Comments to the Author

Reviewer #1: Peer Review for Manuscript PONE-D-25-09667R1

Date: 10-June-2025

“Machine learning for predicting the diagnosis of tuberculous versus malignant pleuraleffusion: external validation and accuracy in two different settings”

The revision mauniscript is much improved. All the suggested points have been resolved. I agree that the paper is published.

My general comments:

In general, the authors presented a well-conducted of external validation study. This research was developed, based on a authors’ recent publication of development prospective cohort model (reported in 2021) of Adenosine deaminase (ADA) for pleural tuberculosis in low tuberculosis (TB) prevalence. The training cohort showed promising results, providing a good rationale for an external validation study. The analysis is very good; however, I think there is room for improvement in writing. To my own perception, I think the structure and presentation of this article writing are still not smooth.

The plus point of this external validation is that the developed pleural TB model (TPE) was tested in a completely different cohort with different TB/TPE incidence. In this study, although test cohort showed a bit lower accuracy, predictive values, than the Train study cohort. I think these real-life data are highly appreciated, demonstrating the applicability of the machine learning TPE models. This highlights the real-life data testing in the external validation cohort.

In clinical practice, tuberculous pleural effusion (TPE) is difficult for diagnosis, particularly in immunocompromised patients and/or limited resource countries with high TB burden. Hence, I raise a clinical question regarding the immune status of the study participants (training and test cohorts). Because HIV co-infection prevalence is highly significant among patients with TPE, do authors have any data about HIV infection or immune testing (CD4 cell counts…etc) in these cohorts?

In additionally, a short justification for sample size calculation in external validation cohorts enhances the validity of the study as well.

My review: All my comments were appropriately answered.

My specific comments:

1. Title: The title fully describes study aim and objectives.

My review: The updated title is accepted.

2. Abstract: The abstract is well written. However, I also think one minor point for amendment, as below:

Lines 45 to 47:

The authors presented a sample size of the Test cohort with 832 consecutive patients in Bajo Deba health district (1996-2012), but did not show the sample size of the Train cohort from a prospective cohort study (2013-2020). To be consistent and transparent in data, the sample size (how many patients in Train cohort?) should be described (stated) in this section as well.

My review: These comments have been amended as appropriate.

3. Introduction:

There is room for improvement in the introduction section, as follows:

Line 67: 95% UI needs to be fully written, as readers are not familiar with this term-UI.

My review: My comment was resolved as appropriate.

Lines 92 to 94: “The model is freely 92 available as an app (at https://pleurapp.ispana.es/) to help other physicians or thoracic surgeons apply this approach when dealing with exudative and lymphocytic pleural effusions”

This information should not be placed in the introduction because it was not connected with the flow of the main idea discussed. This can be relocated to the discussion as appropriate.

My review: My comment was resolved as appropriate.

Lines 99 to 100 in the introduction section:

“we compared the 99 diagnostic accuracy of the ML procedure and the classical Bayesian analysis system for TPE in both 100 different clinical scenarios (Bajo Deba 1996-2012 and Gipuzkoa 2013-2022)”.

The authors mentioned two study cohorts to be modelled without brief introduction before. Hence, I recommend a brief introduction (1-2 sentences) of these 2 cohorts in the previous paragraphs.

My review: My comment was resolved as appropriate.

4. Materials and Methods and Results

There is room for Materials and Methods section for improvement, regarding data presentation.

4.1 I can understand that the authors aimed to place emphasis on the Test cohort (external validation), so proactively present the Test study cohort as the first group, while the Train study cohort as the second group. Intuitively, this style of data presentation brings readers (like me) to a certain level of confusion and needs to reread and rethink. Therefore, I recommend that the authors present data as routine to characterize the first group = Train cohort, and second group = external validation cohort.

This order of presentation should be consistent throughout the manuscript text and Tables. In the Tables 1 and 2, the authors first present: Test to Train cohorts (in sequence);

then in Tables 3 and 4: Train to Test cohorts (in sequence)

The transition in data presentation will make readers (eg, like my case) confused and take some time to reread, rethink the study data.

Therefore, I highly recommend a consistent presentation style of study data, I prefer training to testing cohort data presentation (from left to right), as the conventional way.

My review: My comment was resolved as appropriate.

4.2 In Lines 213 to 218, the authors do not need to describe detailed data about confusion matrices of all machine learning models one by one, because readers can track all these information in Figures 2 and 4 presented. Only salient features from these data should be stated in the manuscript text.

My review: My comment was resolved as appropriate.

4.3 Lines 237-239 and Lines 242-245

“The estimated PPV area is calculated as a function of the pre-test probability (prevalence) using the

sensitivity and specificity of each classifier obtained in the training dataset (Gipuzkoa):

(sensitivity * prevalence) / ((sensitivity * prevalence) + (1 - specificity) * (1 - prevalence)).”

“The real PPVs in Gipuzkoa and Bajo Deba are calculated as the true positives divided by the sum of the true positives and false positives, obtained in each dataset: TP / (TP + FP)”

The formulas should be relocated in the Methods section, as it is more appropriate.

My review: My comment was resolved as appropriate.

5. Discussion:

The discussion is comprehensively discussed and well written. From this study, the machine learning models outperformed the Bayesian modelling, as shown in a different study setting with different prevalence of TPE and malignancy.

My review: My comment was resolved as appropriate.

Conclusion:

I think this is a great study. The revised manuscript resolved all my comments. I agree that this study is appropriate for publication.

Many thanks,

Best regards,

Reviewer #3: Authors have addressed the revisions. All required questions have been answered and that all responses meet formatting specifications

7. PLOS authors have the option to publish the peer review history of their article (what does this mean? ). If published, this will include your full peer review and any attached files.

**Do you want your identity to be public for this peer review?** For information about this choice, including consent withdrawal, please see our Privacy Policy .

Reviewer #1: **Yes: ** Nguyen Tat Thanh, MD PhD

Reviewer #3: **Yes: ** Harun Agca

---

## [Editor Report · Acceptance letter]

PONE-D-25-09667R1

PLOS ONE

Dear Dr. Garcia-Zamalloa,

I'm pleased to inform you that your manuscript has been deemed suitable for publication in PLOS ONE. Congratulations! Your manuscript is now being handed over to our production team.

Kind regards,

on behalf of

Dr. Guocan Yu

Academic Editor

PLOS ONE